# PRMT5 Mediated HIF1α Signaling and Ras-Related Nuclear Protein as Promising Biomarker in Hepatocellular Carcinoma

**DOI:** 10.3390/biology13040216

**Published:** 2024-03-27

**Authors:** Wafaa Abumustafa, Darko Castven, Fatemeh Saheb Sharif-Askari, Batoul Abi Zamer, Mawieh Hamad, Jens-Uwe Marquardt, Jibran Sualeh Muhammad

**Affiliations:** 1Department of Basic Medical Sciences, College of Medicine, University of Sharjah, Sharjah 27272, United Arab Emirates; 2Research Institute of Medical and Health Sciences, College of Medicine, University of Sharjah, Sharjah 27272, United Arab Emirates; 3First Medical Department, University Medical Center Schleswig-Holstein, Campus Lübeck, 23538 Lübeck, Germany; 4Department of Pharmacy Practice and Pharmacotherapeutics, College of Pharmacy, University of Sharjah, Sharjah 27272, United Arab Emirates; 5Department of Medical Laboratory Sciences, College of Health Sciences, University of Sharjah, Sharjah 27272, United Arab Emirates

**Keywords:** protein arginine N-methyltransferase 5, HIF1α, hepatocellular carcinoma, Ras-related nuclear protein

## Abstract

**Simple Summary:**

The protein arginine N-methyltransferase 5 (PRMT5) has been identified as a promising therapeutic target in various cancers. However, its role in hepatocellular carcinoma (HCC) development has not yet been investigated. This study aims to understand PRMT5′s impact on overall survival, signaling pathways, and downstream gene expression using in silico public databases and our in-house NGS data. Our results revealed an increase in *PRMT5* expression in HCC compared to normal liver tissue, and this elevated expression was associated with poorer patient outcomes. Analysis of promoter CpG islands and methylation status suggested potential epigenetic mechanisms driving *PRMT5* overexpression in HCC. Pathway analyses found a link between *PRMT5* expression and the HIF1α pathway, with Ras-related nuclear protein (RAN) identified as a potential target of PRMT5 in HCC.

**Abstract:**

Protein arginine N-methyltransferase 5 (PRMT5) has been identified as a potential therapeutic target for various cancer types. However, its role in regulating the hepatocellular carcinoma (HCC) transcriptome remains poorly understood. In this study, publicly available databases were employed to investigate PRMT5 expression, its correlation with overall survival, targeted pathways, and genes of interest in HCC. Additionally, we utilized in-house generated NGS data to explore *PRMT5* expression in dysplastic nodules compared to hepatocellular carcinoma. Our findings revealed that PRMT5 is significantly overexpressed in HCC compared to normal liver, and elevated expression correlates with poor overall survival. To gain insights into the mechanism driving *PRMT5* overexpression in HCC, we analyzed promoter CpG islands and methylation status in HCC compared to normal tissues. Pathway analysis of *PRMT5* knockdown in the HCC cells revealed a connection between *PRMT5* expression and genes related to the HIF1α pathway. Additionally, by filtering PRMT5-correlated genes within the HIF1α pathway and selecting up/downregulated genes in HCC patients, we identified Ras-related nuclear protein (RAN) as a target associated with overall survival. For the first time, we report that PRMT5 is implicated in the regulation of *HIF1A* and *RAN* genes, suggesting the potential prognostic utility of PRMT5 in HCC.

## 1. Introduction

According to the records of the International Agency for Research on Cancer (IARC), in 2020, primary liver cancer was ranked sixth in incidence and third in mortality rate. The vast majority, approximately 80%, of primary liver cancer cases are hepatocellular carcinoma (HCC) [1]. Several risk factors have been associated with HCC, including metabolic liver diseases, such as nonalcoholic fatty liver (NAFL) and nonalcoholic steatohepatitis (NASH), chronic viral infections, such as hepatitis B virus (HBV) and hepatitis C virus (HCV), alcohol abuse, inherited diseases, and Aflatoxin exposure [2]. HCC is classified into five stages, and treatment is determined based on the stage. It begins with resection and liver transplantation in the early stages, followed by targeted therapy (such as Sorafenib) in the advanced stages [3,4]. Most of the early-stage cases achieve a 5-year survival rate, and 20–50% of intermediate-stage cases attain a 3-year survival rate. Unfortunately, terminal-stage patients succumb to the disease within 6 months [5,6]. Therefore, identifying new diagnostic biomarkers, therapeutic targets, and prognostic markers is critical for diagnosing and treating the disease at its early-intermediate stages.

Recent studies have suggested that protein arginine N-methyltransferase 5 (PRMT5) could serve as a potential prognostic and therapeutic biomarker in various cancers, including breast, lung, and colorectal cancer [7,8,9]. PRMT5 functions as a catalytic enzyme that transfers methyl groups from S-adenosylmethionine (SAM) to arginine residues in multiple proteins, including histones [10]. Arginine monomethylation or symmetric dimethylation induced by PRMT5 affects cellular functions by influencing protein activity and stability, gene expression, and pre-mRNA splicing [11,12]. There is also evidence suggesting that PRMT5 plays a role in promoting cancer cell growth and proliferation [9,13]. Interestingly, PRMT5 overexpression has been positively correlated with cellular transformation in various neoplasms, including HCC [14,15,16]. As a potential molecular target, the PRMT5 inhibitor (GSK3326595) is currently in phase I/II clinical trials for acute myeloid leukemia (AML) and other cancer types [17,18]. Also, several studies have established PRMT5 as a reliable prognostic marker for cancers, including HCC [19]. Therefore, investigating PRMT5 expression at different stages of HCC could provide a hint of whether PRMT5 could also serve as a diagnostic marker of early HCC. Moreover, PRMT5 overexpression in the liver has been linked to a high-fat diet, which is a risk factor for HCC [20,21].

Previous research documents the role of PRMT5 in regulating key signaling pathways in HCC, such as the WNT signaling pathway, the ERK signaling pathway, and iron homeostasis. For instance, PRMT5 manipulates WNT signaling activity, which enhances HCC metastases. PRMT5 activates the ERK signaling pathway, which hinders the expression of B-cell translocation gene 2 (BTG2), which is responsible for G1 to S phase cell cycle arrest. On the other hand, PRMT5 exhibited a protective role in HCC via inhibiting ferritin heavy-chain-1 (FTH1) expression which reverses the iron overload process in HCC [22,23,24,25]. However, the role of PRMT5 in regulating the Hypoxia-inducible factor 1 alpha (HIF1α) signaling pathway in HCC has not been previously reported. The HIF1α pathway is involved in HCC proliferation, invasion, metastasis, angiogenesis, and drug resistance [26]. HIF1α is a transcription factor that gets activated under hypoxia condition which is associated with tumor microenvironment. It plays a crucial role in cancer cell survival by inhibiting the generation and propagation of reactive oxygen species (ROS) and by blocking ROS-mediated apoptosis [27,28,29]. HIF1α pathway inhibition demonstrates a promising therapeutic strategy for combating HCC progression and enhancing patient outcomes.

In this study, publicly available data from The Cancer Genome Atlas (TCGA) and Gene Expression Omnibus (GEO) were used to investigate the possible role of PRMT5 in regulating the expression of key genes in the HIF1α pathway. Furthermore, GEO data were utilized to assess the capacity of PRMT5 downstream effectors from the HIF1α pathway to serve as prognostic biomarkers. Our data proposed promoter hypermethylation as a mechanism that is involved in PRMT5 increased expression in HCC. Our data revealed an association between PRMT5 upregulation and HIF1α pathway activation, as well as increased *HIF1A* gene expression. Also, our data highlight the capacity of PRMT5 in regulating *RAN* gene expression, which is a part of the HIF1α pathway and could serve as a biomarker for HCC prognosis.

## 2. Materials and Methods

### 2.1. In Silico Analysis Using UALCAN Online Portal

The UALCAN in silico tool provides access to Level 3 RNA-seq. The UALCAN website (http://ualcan.path.uab.edu/analysis.html, accessed on 29 October 2022) was used to profile gene expression in 371 HCC patients compared to 50 normal counterparts using Cancer Genome Atlas (TCGA) level 3 RNAseq and clinical data [30,31]. In addition, UALCAN was utilized to obtain protein expression analysis via the Clinical Proteomic Tumor Analysis Consortium (CPTAC) and the International Cancer Proteogenomic Consortium (ICPC) datasets that represent High-throughput mass spectrometry data of 165 normal versus 165 HCC patients. The website conducts a Comprehensive Perl Archive Network (CPAN) to calculate the *p*-value using a Student *t*-test.

### 2.2. Methylation Status of Promoter CpG Islands

The genome data viewer browser allowed the visualization of biological information blotted onto the genome in a graph such as CpG islands. A genome data viewer browser provided by the National Health Institute was utilized to identify the presence of CpG islands in the promoter of the gene. The UALCAN tool provides promoter region hypermethylation analysis utilizing TCGA data obtained via Illumina Infinium HumanMethylation450 BeadChip. Promoter methylation level was evaluated in the UALCAN tool using TCGA DNA methylation data of 50 normal compared to 377 HCC cases. The website conducts CPAN to calculate the *p*-value using a Student *t*-test.

### 2.3. Kaplan–Meier Patient Survival Analysis

The Kaplan–Meier plotter (https://kmplot.com/analysis/, accessed on 15 December 2022) was used to analyze RNAseq data to identify overall survival (OS), relapse-free survival (RFS), progression-free survival (PFS), and disease-specific survival (DSS) in 364, 316, 370, and 362 liver cancer patients, respectively [32,33]. A upper quartile cutoff was used to generate a *PRMT5* Kaplan–Meier plotter graph of OS, RFS, PFS, and DSS. A lower quartile cutoff was used to generate the *RAN* Kaplan-Meier plotter graph of OS, RFS, PFS, and DSS. A hazard ratio (HR) of more than 1 was considered as a bad prognosis biomarker, while an HR of less than 1 was considered as a good prognosis. An upper quartile cutoff was used to generate a *MAPK3* Kaplan–Meier plotter graph of OS, RFS, PFS, and DSS. Auto select option for the best cutoff value was used to generate an *FLT1* and *SERPINE1* Kaplan–Meier plotter graph of OS.

### 2.4. QIAGEN Ingenuity Pathway Analysis (IPA)

Transcriptome data generated by Illumina Novaseq 6000 for PRMT5 knocked down JHH-7 Cell utilizing a short hairpin RNA (shRNA) were obtained from the Gene Expression Omnibus (GEO) database (GSE168745) provided by the National Center for Biotechnology Information (https://www.ncbi.nlm.nih.gov/geo/, accessed on 30 November 2022). Significantly altered gene lists were utilized to perform the IPA using the tool provided by Qiagen to analyze omics data.

### 2.5. TIMER 2.0

The Tumor Immune Estimation Resource (TIMER) is a database that represents a molecular cross-talk of tumor and tumor microenvironment. TIMER also allows the detection of gene expression correlation in multiple cancers. TIMER (https://cistrome.shinyapps.io/timer/, accessed on 9 January 2023) was used to investigate gene–gene correlations in 371 specimens of liver hepatocellular carcinoma sourced from Cancer Genome Atlas (TCGA) [34,35]. Spearman’s rho value was utilized to identify the degree of the correlation between two genes. An R-value between 0 and 0.3 was considered weak positive, 0.3 to 0.7 moderate positive, and 0.7 to 1 strong positive.

### 2.6. Next-Generation Sequencing (NGS)

RNA was isolated from 12 samples of dysplastic nodules and 7 samples of HCC tumor tissue. Library preparation was performed using the NEBNext^®^ UltraTM RNA Library Prep Kit for Illumina^®^, and sequencing was conducted using Illumina Novaseq 6000. The data were aligned to the human reference genome sequence (ENSEMBL Homosapiens.GRCh38) using HISAT2 (hisat2-2.0.2-beta). These data were generated by our co-authors and were previously published [36]. The Bioconductor limma-voom package was utilized for RNA-seq data normalization.

### 2.7. Statistical Analysis

GraphPad Prism 8.4.2 was used to perform an unpaired parametric *t*-test to calculate the *p*-value. Collective data were presented as mean ± SEM. A *p*-value of <0.05 was considered to be statistically significant. The SPSS program was used to identify the distribution of the dataset and to draw the receiver operating characteristic (ROC) curve utilizing the GEO data set (GSE214846) to assess the gene expression clinical value as a diagnostic marker.

## 3. Results

### 3.1. PRMT5 Is Over-Expressed in HCC and Differentially Expressed in Different Disease Stages

To identify PRMT5 expression in HCC, the UALCAN tool was used to analyze TCGA RNAseq, Clinical Proteomic Tumor Analysis Consortium (CPTAC), and the International Cancer Proteogenome Consortium (ICPC) data. In silico analysis showed that PRMT5 is overexpressed in HCC at the RNA and protein levels with *p* < 0.0001 (Figure 1a,b). *PRMT5* mRNA was significantly overexpressed in all stages of HCC relative to its normal counterpart; it was very highly elevated in stages II (*p* < 0.05) and III (*p* < 0.01) relative to stage I (Figure 1c). Additionally, transcriptomic data demonstrated a significant increase in the *PRMT5* expression level of HCC compared to dysplastic nodules with *p* < 0.01 (Figure 1d), indicating a progressive activation of PRMT5 during liver cancer development and progression.

### 3.2. PRMT5 Is a Promising Disease Progression Marker for HCC

We next evaluated the correlation between *PRMT5* expression and patient survival. High expression of *PRMT5* significantly correlated with poor overall survival *p* < 0.05, relapse-free survival with *p* < 0.05, and progression-free survival with *p* < 0.05, except disease-specific survival with *p* > 0.05 in HCC patients (Figure 2a–d). The capacity of PRMT5 to serve as a disease progression marker ROC curve was also evaluated by comparing HCC versus normal adjacent tissues using the GSE214846 dataset. This type of analysis revealed that the area under the curve is 81.5%, sensitivity 73.8%, and specificity 89.2%, with a cutoff value of 5 (Figure 2e). This highlights the capability of PRMT5 to serve as a biomarker in HCC.

### 3.3. PRMT5 Promoter Is Hypomethylated in HCC

To confirm the presence of CpG islands in the promoter region of the *PRMT5* gene, the Genome data viewer browser was used. An 872-nucleotide long cytosine- and guanine-rich domain present at NC_000014.9 [22,929,030–22,929,901] chromosome 14 that contains promoter region of *PRMT5* gene was identified (Figure 3a). *PRMT5* promoter methylation status was assessed using the publicly available tool. The results showed a significant hypomethylation at the promoter region of *PRMT5* in HCC (*p* < 0.0001) patients compared to normal counterparts, especially in early HCC stages (*p* < 0.0001) (Figure 3b,c). Indicating the role of epigenetic hypomethylation of *PRMT5* promoter region in *PRMT5* overexpression in the early stage of HCC.

### 3.4. PRMT5 Knockdown Manipulates the Activity of the HIF1α Pathway

To investigate the pathways related to *PRMT5* expression, transcriptomic analysis of the *PRMT5*-silenced JHH-7 HCC cell line was performed using IPA (−log (*p*-value) = 6.6). *PRMT5* depletion significantly reduced the expression of genes involved in the HIF-1α pathway (Figure 4a). In that, *HIF1A* gene expression showed a moderate positive correlation with *PRMT5* expression in HCC patients with *p* < 0.0001 (Figure 4b). The detailed IPA visual pathway is shown in Appendix A, showing that *PRMT5* silencing helped in the deactivation of HIF-1α via reduced expression of key genes such as MAPK3 (ERK1/2) and ribosomal S6 kinase B2 (RPS6KB2 or p70S6Kb). Also, inhibition of angiogenesis and blood vessel maturation pathways was evident as a decreased expression of the vascular endothelial growth factor receptor 1 (FLT1) and the plasminogen activator inhibitor-1 (SERPINE1) (Appendix A). PRMT5 showed a significant (*p* < 0.0001) but weak positive (r < 0.3) correlation with FLT1 and SERPINE1 (Appendix A). It is worth noting that FLT1 and SERPINE1 expression were not significantly associated with overall survival according to Kaplan–Meier patient survival analysis (Appendix A). *PRMT5* inhibition also reduced the expression of Ras-related nuclear protein (RAN) (Appendix A). Collectively, these data indicate that PRMT5 enhances the expression of genes involved in tumor progression via HIF-1α pathway activation.

### 3.5. PRMT5 Expression Is Positively Correlated with MAPK3, and RAN Genes as a Key Regulator in HCC

To further understand the relationship between PRMT5 and the HIF1α pathway, the possibility that PRMT5 could be involved in the regulation of the key HIF1α pathway was investigated. In silico analysis was performed to address this point, with *RAN* and *MAPK3* as the genes that were significantly co-expressed with PRMT5 in HCC patients, the genes that correlated with worse prognosis, and genes that were downregulated upon PRMT5 depletion were shortlisted (Appendix A). The *MAPK3* gene was significantly upregulated in HCC patients with early, intermediate, and late stages of the disease (*p* < 0.0001) (Appendix A). Also, a moderate positive correlation between *PRMT5* and *MAPK3* was observed in HCC (*p* < 0.0001) (Appendix A). *MAPK3* overexpression was significantly associated with HCC patient overall survival (OS) (*p* < 0.001) and disease-specific survival (DSS) (*p* < 0.05). However, it is not linked to relapse-free survival (RFS) (*p* > 0.05) or progression-free survival (PFS) (*p* < 0.05) (Appendix A).

Further analysis showed increased expression of the RAS-related nuclear protein (RAN) in HCC clinical samples both at the transcriptomic and protein levels (*p* < 0.0001) (Figure 5a,b). TCGA data showed a higher expression of *RAN* in stages 2 (*p* < 0.05) and 3 (*p* < 0.001) than in stage 1 (Figure 5c). A significant upregulation of *RAN* gene expression in HCC tissue versus dysplastic nodule was detected using NGS data (*p* < 0.05) (Figure 5d). Furthermore, a moderate positive correlation between *PRMT5* and *RAN* was detected in HCC patients (Figure 5e). To investigate the prognostic and diagnostic value of *RAN* gene expression, a survival curve, and a ROC curve analysis were performed. *RAN* overexpression was remarkably linked to poor overall survival (OS), relapse-free survival (RFS), progression-free survival (PFS), and disease-specific survival (DSS) in HCC patients (Figure 6a–d). ROC analysis of the GSE214846 dataset showed that the area under the curve is 86.2%, and both sensitivity and specificity are 86.2% with a cutoff of 65 (Figure 6e).

## 4. Discussion

Our analysis of the data has unveiled a notable pattern of PRMT5 overexpression, specifically in hepatocellular carcinoma (HCC). This overexpression is not uniform but exhibits a gradual increase as the disease progresses through various stages. HCC’s progression is well-documented to follow a stepwise sequence, initiating from regenerative nodules within the liver parenchyma and gradually evolving into dysplastic nodules (DNs). These DNs are known to possess cellular abnormalities that indicate a higher risk of developing into fully formed HCC lesions. Our findings highlighted the distinct elevation of PRMT5 levels in HCC compared to dysplastic nodules, indicating a significant role for PRMT5 in the step-by-step advancement of HCC. This observation underscores the potential of PRMT5 as a key player in the intricate process of HCC development. The gradual increase in PRMT5 expression across different stages of HCC progression suggests a dynamic involvement of PRMT5 in driving the malignant transformation of liver cells. Moreover, understanding this progressive upregulation of PRMT5 in HCC sheds light on the molecular mechanisms underlying the transition from pre-neoplastic lesions to fully established HCC. This insight is crucial for identifying potential therapeutic targets aimed at disrupting the pathways influenced by PRMT5, thereby impeding or reversing the progression of HCC. Further investigations into the specific interactions and downstream effects of PRMT5 in HCC development are warranted to fully comprehend its role and therapeutic implications in combating this aggressive form of cancer.

While previous studies have explored the role of PRMT5 in HCC, transcriptional activation of PRMT5 in HCC has not been examined. Multiple transcription factors have been implicated in PRMT5 overexpression in different cancer subtypes, including nuclear transcription factor Y (NF-Y) in prostate cancer, nuclear factor kappa B (NF-κB) in diffuse large B-cell lymphoma, and the fused MLL-1 protein in acute myeloid leukemia (AML) [37,38,39]. Additionally, epigenetic alterations, such as N-alpha-acetyltransferase 40 (NAA40)-induced acetylation, have been reported to contribute to PRMT5 overexpression in colorectal cancer [40]. DNA methylation, a heritable epigenetic alteration, occurs due to DNA methyltransferase (DNMTs) activity adding a methyl group to cytosine [41]. DNA methylation plays a crucial role in gene expression modulation by hindering the binding of transcription factors to DNA [42]. Our study suggests DNA hypomethylation as a potential cause of PRMT5 overexpression in the early stage of HCC. However, other mechanisms could influence PRMT5 expression in later stages, such as histone modifications and transcription factors. Indeed, identifying PRMT5 promoter CpG islands hypermethylation in HCC cell lines in vitro is crucial in addition to validation in patient biopsy samples.

The HIF1α pathway is a critical signaling pathway involved in the development and progression of HCC [43,44]. Under hypoxic conditions, HIF1α stabilizes and translocates to the nucleus, where it induces the expression of genes involved in glucose metabolism, angiogenesis, and cell proliferation [45]. HIF1α is often overexpressed in HCC, promoting tumor growth, metastasis, and drug resistance, with increased expression associated with poor patient outcomes [26]. Our data shed light on a positive relationship between PRMT5 and HIF1α expression as well as HIF1α pathway activity. Even though some of the previous work proposed a protective role of PRMT5 in HCC, our data indicated inhibition of HIF1α signaling upon targeting PRMT5. Therefore, we emphasize the previous data that identify PRMT5 as a promising therapeutic target in HCC [22,23,24,25]. Indeed, testing the therapeutic capacity of PRMT5 inhibitors against HCC via clinical trials is pivotal.

Furthermore, we identified the role of PRMT5 in regulating the expression of potential biomarker genes involved in the HIF1α pathway. Our results revealed a correlation between PRMT5 expression and two such biomarkers, MAPK3 and RAN. Previous studies discussed PRMT5′s role in modulating the activity of ERK in glioblastoma neurospheres and lung cancer, with ERK being encoded by the *MAPK3* gene [46,47]. Our study also showed a positive correlation between *PRMT5* and *MAPK3* gene expression in HCC patients and the JHH-7 HCC cell line.

RAN is a small GTP-binding protein crucial for RNA and protein transport through the nuclear membrane [48]. It plays a role in microtubule polymerization and mitotic spindle formation, thus impacting the cell cycle [49]. RAN has been implicated in tumor progression and metastasis in various cancer subtypes, such as breast and pancreatic cancer, where it affects proliferation and apoptosis [49,50]. RAN has also been considered a potential therapeutic target in certain cancers [51]. Its expression in cancer has been linked to epigenetic regulation, such as long noncoding RNA LINC00858 in gastric cancer and microRNAs (MiR-802) in colorectal cancer [52,53]. Furthermore, *RAN* expression has been correlated with promoter hypomethylation in HCC and suggested as a potential prognostic marker [54]. Our data supported these findings, indicating *RAN* as a prognostic biomarker. This study establishes a positive correlation between *PRMT5* and the *RAN* gene through co-expression analysis in HCC patients and investigation of *RAN* expression upon *PRMT5* knockdown in the HCC cell line, suggesting a regulatory effect of PRMT5 on *RAN*. It is worth investigating whether the PRMT5 regulatory effect is due to epigenetic modulation, where it is capable of inducing histone symmetric demethylation, or due to alternative splicing. Chromatin immunoprecipitation assay targeting PRMT5 unique histone modifications such as H4R3me2s and H3R8me2s and RNAseq-based alternative splicing analysis could expand the knowledge of *PRMT5*-induced expression regulation of *RAN* [9]. We propose *RAN* and *PRMT5* as promising disease progression biomarkers for HCC, with higher specificity (>80%) compared to alpha-fetoprotein (AFP), which has a histological specificity of 70.4% [55]. Immunohistochemistry screening of RAN and PRMT5 expression in HCC tissue is warranted, along with further investigations involving a larger sample size population. Additionally, we illustrate the significant difference in *RAN* expression between DN and HCC and report variable expression across different disease stages. Indeed, expanding the experiment into a larger sample size would allow further generalization of our hypothesis. RAN translocate HIF1α into the cytoplasm, contributing to cancer progression in mitochondria [27,28,29]. Understanding the mechanism of action in which RAN contributes to HCC pathogenesis will strengthen the knowledge and pave the way for further research in this field. Therefore, PRMT5 and RAN could be a promising diagnostic marker for early-stage HCC and allow early detection of the disease, which is correlated with better patient outcomes.

## 5. Conclusions

In conclusion, the data presented in this study strongly suggested a positive correlation between PRMT5 expression and various pro-cancer pathways, notably the HIF1α pathway in HCC. The intricate analysis conducted revealed compelling evidence indicating that PRMT5 may play a role in activating the HIF1α pathway, potentially through the MAPK3 (ERK1/2) signaling cascade. This activation mechanism could significantly contribute to the enhanced growth and metastatic potential observed in HCC cells with elevated PRMT5 expression levels. Furthermore, our findings point towards a directly proportional relationship between PRMT5 and the *RAN* gene, a pivotal component within the HIF1α pathway. The identification of RAN as a potential diagnostic biomarker underscores the clinical relevance of understanding PRMT5′s involvement in HCC pathogenesis. However, it is essential to acknowledge that our study represents a stepping stone in unraveling the complex molecular interactions underlying the PRMT5-HIF1α pathway axis in HCC. Further in-depth research is warranted to fully elucidate the precise mechanisms through which PRMT5 activates the HIF1α pathway and to decipher the functional consequences of this activation. These insights are crucial for developing targeted therapeutic strategies aimed at disrupting the PRMT5-driven pathways implicated in HCC progression, ultimately improving patient outcomes and treatment efficacy in combating this aggressive form of liver cancer. 

## Figures and Tables

**Figure 1 biology-13-00216-f001:**
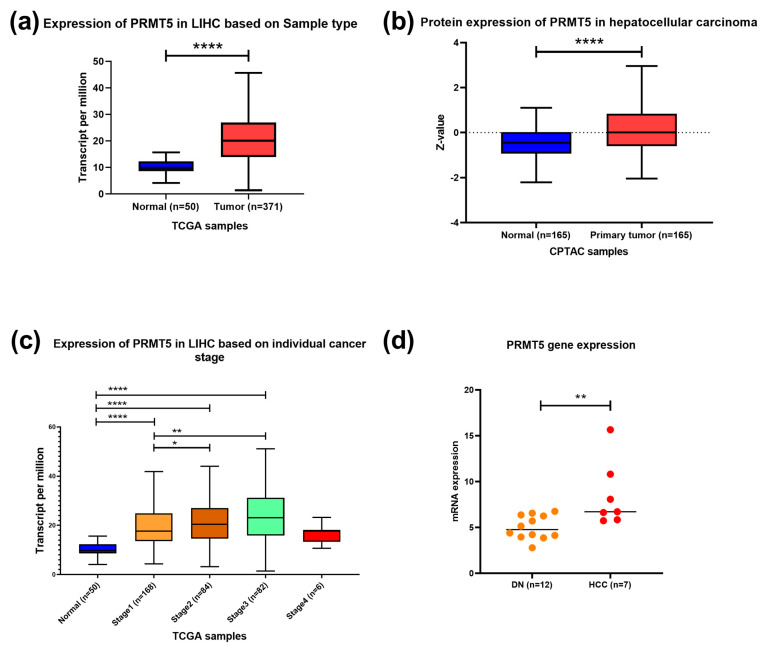
PRMT5 expression in HCC. (**a**) *PRMT5* expression in HCC clinical samples compared to normal tissue counterpart using RNA seq dataset, TCGA data, UALCAN tool. (**b**) PRMT5 expression in HCC clinical samples compared to normal tissue counterpart using clinical proteomic tumor analysis consortium (CPTAC) data, UALCAN tool. (**c**) *PRMT5* expression in different stages of HCC compared to normal tissue counterpart using RNA seq dataset from TCGA data, UALCAN tool. (**d**) transcriptome data of *PRMT5* expression in 12 samples of dysplastic nodules and 7 samples of hepatocellular carcinoma. * *p* < 0.05, ** *p* < 0.01, and **** *p* < 0.0001.

**Figure 2 biology-13-00216-f002:**
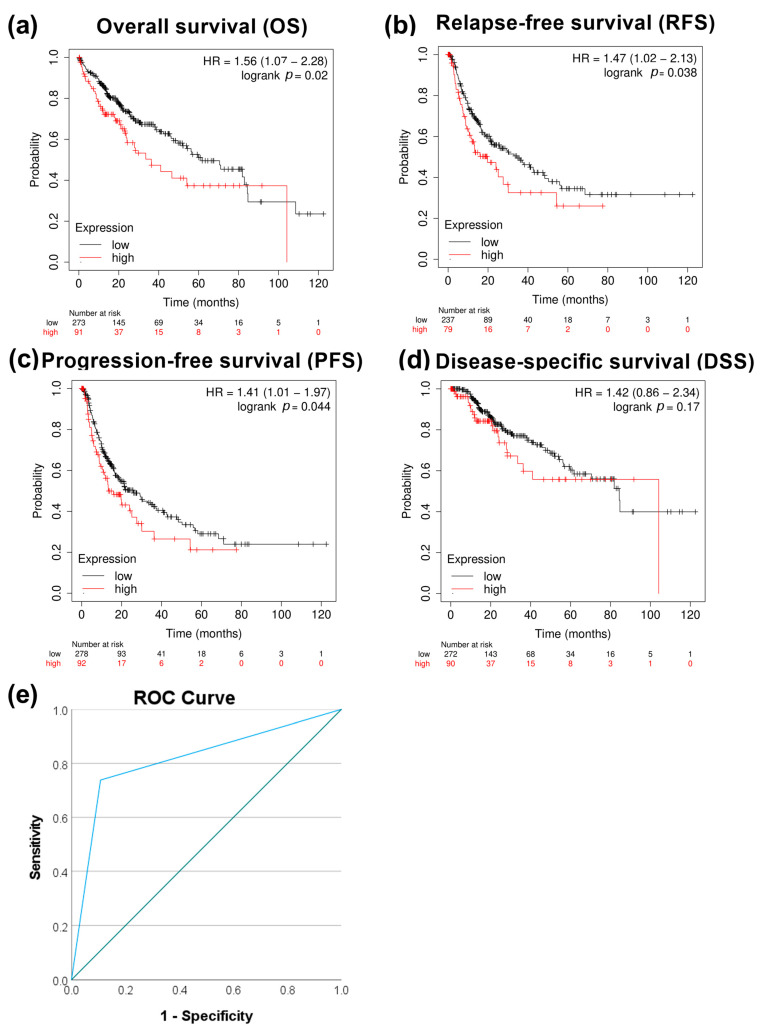
Correlational analysis of *PRMT5* expression versus HCC patient survival. (**a**) *PRMT5* expression versus overall patient survival (OS). (**b**) *PRMT5* expression versus relapse-free survival (RFS). (**c**) *PRMT5* expression versus progression-free survival (PFS). (**d**) *PRMT5* expression versus disease-specific survival (DSS). (**e**) ROC curves for *PRMT5* expression at adjacent normal versus HCC tissue utilizing the GSE214846 dataset.

**Figure 3 biology-13-00216-f003:**
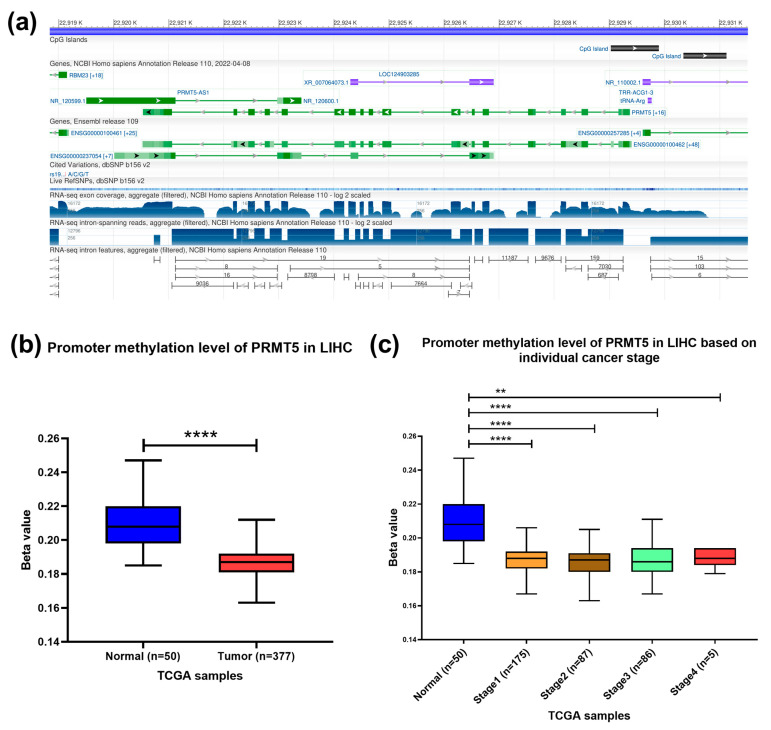
Methylation level of *PRMT5* promoter CpG Islands in HCC. (**a**) Location of CpG islands in the promoter region of *PRMT5* gene. (**b**) *PRMT5* promoter methylation level in HCC compared to normal tissues counterpart as per the TCGA database, UALCAN tool. (**c**) *PRMT5* promoter methylation level at different stages of HCC compared to normal tissue counterpart as per the TCGA database, UALCAN tool., ** *p* < 0.01, and **** *p* < 0.0001.

**Figure 4 biology-13-00216-f004:**
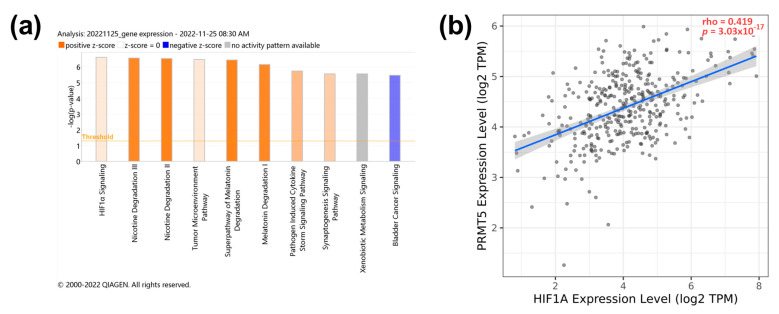
PRMT5 expression positively correlates with HIF1α signaling pathway. (**a**) IPA pathway analysis of PRMT5 knocked down JHH-7. (**b**) Spearman’s rank correlation coefficient of PRMT5 versus HIF1α expression.

**Figure 5 biology-13-00216-f005:**
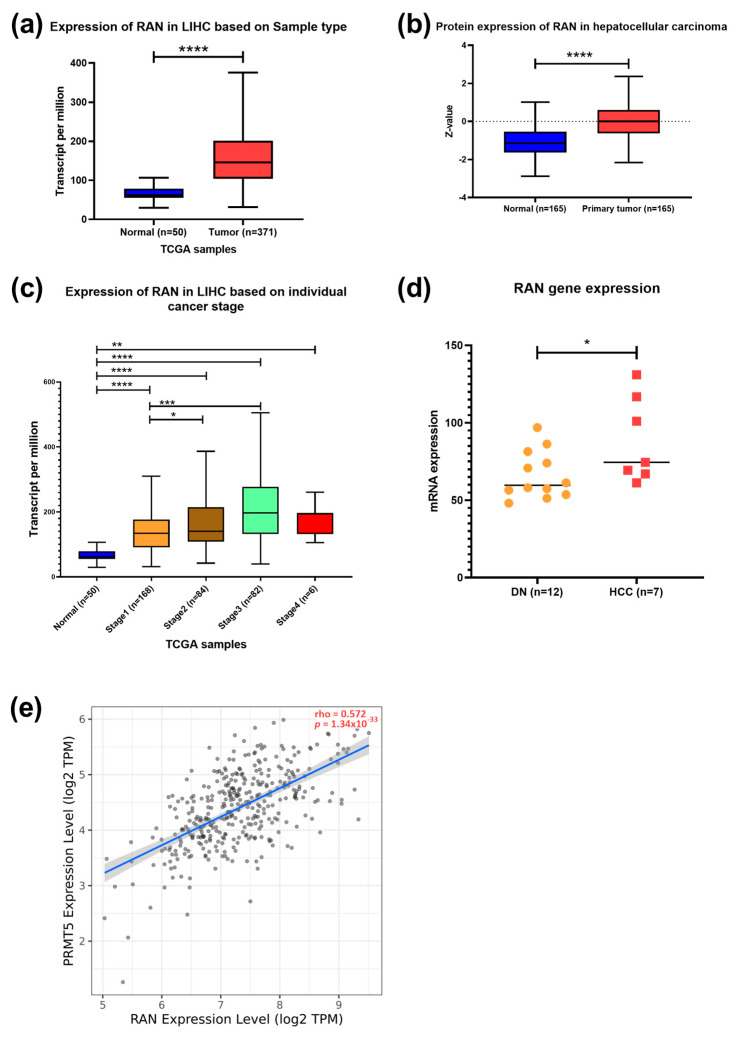
Expression and correlation of *RAN* versus PRMT5 in HCC. (**a**) *RAN* expression in HCC compared to normal tissue counterpart. (**b**) *RAN* expression in different stages of HCC compared to normal tissue counterpart. (**c**) *RAN* expression in different stages of HCC compared to normal tissue counterpart. (**d**) Transcriptomic data of *RAN* expression in 12 samples of dysplastic nodules and 7 samples of hepatocellular carcinoma. (**e**) Spearman’s rank correlation coefficient of *PRMT5* expression and *RAN*. Data in a–c were generated using RNA seq datasets, TCGA database in UALCAN tool; * *p* < 0.05, ** *p* < 0.01, *** *p* < 0.001 and **** *p* < 0.0001.

**Figure 6 biology-13-00216-f006:**
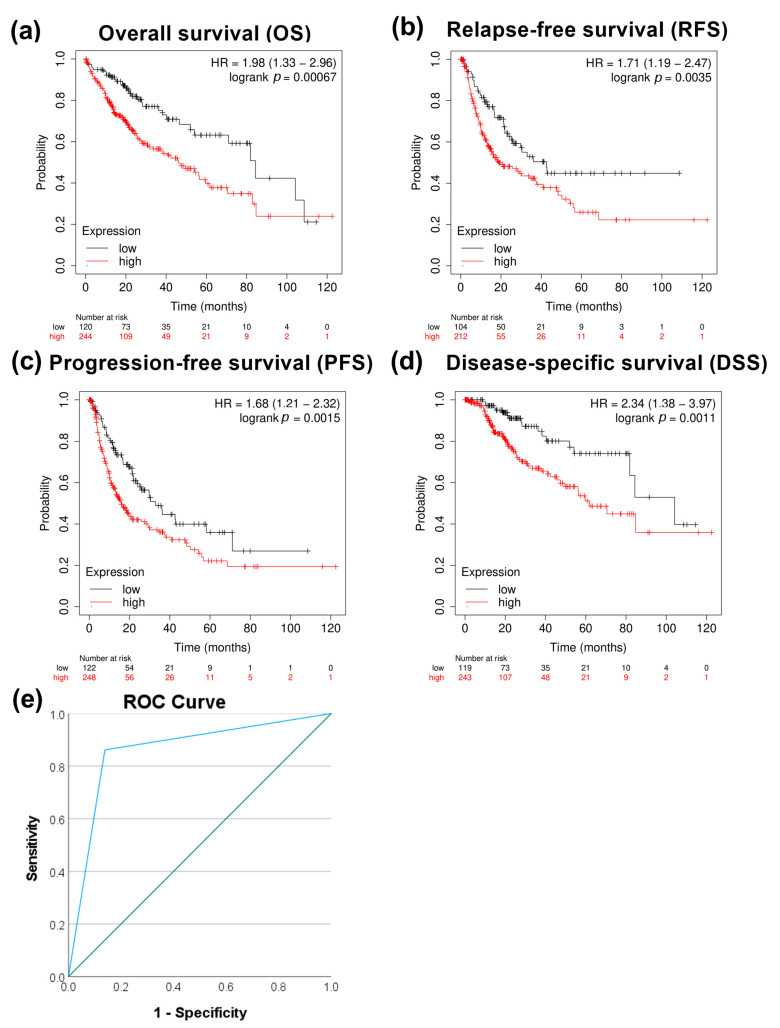
Correlational analysis of *RAN* expression versus HCC patient survival. (**a**) *RAN* expression versus overall survival (OS). (**b**) *RAN* expression versus relapse-free survival (RFS). (**c**) *RAN* expression versus progression-free survival (PFS). (**d**) *RAN* expression versus disease-specific survival (DSS). (**e**) ROC curves for *RAN* expression at adjacent normal versus HCC tissue utilizing the GSE214846 dataset.

## Data Availability

All data in this study were available in the public database. These data can be obtained from the GEO database (GSE168745, and GSE214846).

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
