# Peer review of "PRMT5 Mediated HIF1α Signaling and Ras-Related Nuclear Protein as Promising Biomarker in Hepatocellular Carcinoma"

_biology, 2024, doi:10.3390/biology13040216_

Round 1

Reviewer 1 Report

Comments and Suggestions for Authors

In this study, Abumustafa and colleagues utilized publicly available databases to investigate the correlation between PRMT5 and hepatocellular carcinoma (HCC). Their study revealed that PRMT5 is significantly overexpressed in HCC and that elevated PRMT5 expression is associated with poor survival outcomes. Moreover, their analysis identified a functional link between PRMT5 and HIFα signaling. Specifically, they highlighted the RAN gene as a potential biomarker for HCC.

Comments are as belows:

1. line 106: "This high-lights the capability of PRMT5 to serve as a biomarker in CRC. " I think here should be "HCC" but not "CRC".

2. line 135: "key genes such as MAPK3 (ERK1/2) and ribosomal S6 kinase B2 (RPS6KB2) ". According to the supplementary Fig 1, it should be p70S6K, not RPS6KB2.

3. In HCC, PRMT5 mRNA levels were found to be elevated in stages II and III compared to stage I, as depicted in Figure 1c. However, there appeared to be no significant difference in PRMT5 promoter methylation levels between stages II/III and stage I, as illustrated in Figure 3c. The authors are encouraged to confirm these findings and provide an explanation for the observed elevation in PRMT5 mRNA levels in stages II and III despite the lack of significant differences in promoter methylation levels.

4. Some figures in the manuscript suffer from issues with legibility and quality. Specifically, the fonts are too small, making the figures difficult to read. Such as Figure 1c, 2e, 2f, 3a, 3c, 4a...

Author Response

Reviewer 1

In this study, Abumustafa and colleagues utilized publicly available databases to investigate the correlation between PRMT5 and hepatocellular carcinoma (HCC). Their study revealed that PRMT5 is significantly overexpressed in HCC and that elevated PRMT5 expression is associated with poor survival outcomes. Moreover, their analysis identified a functional link between PRMT5 and HIFα signaling. Specifically, they highlighted the RAN gene as a potential biomarker for HCC.

Comments are as below:

  1. line 106: "This high-lights the capability of PRMT5 to serve as a biomarker in CRC. " I think here should be "HCC" but not "CRC".

Response: The sentence was fixed and currently found line 131 due to other edits

  1. line 135: "key genes such as MAPK3 (ERK1/2) and ribosomal S6 kinase B2 (RPS6KB2) ". According to the supplementary Fig 1, it should be p70S6K, not RPS6KB2.

Response: Supplementary Fig 1 represents some genes by corresponding protein names; in this case, RPS6KB2 is a gene that encodes an enzyme that in humans is termed p70S6Kb. This was clarified in line 162.

  1. In HCC, PRMT5 mRNA levels were found to be elevated in stages II and III compared to stage I, as depicted in Figure 1c. However, there appeared to be no significant difference in PRMT5 promoter methylation levels between stages II/III and stage I, as illustrated in Figure 3c. The authors are encouraged to confirm these findings and provide an explanation for the observed elevation in PRMT5 mRNA levels in stages II and III despite the lack of significant differences in promoter methylation levels.

Response: We suggest that promoter hypomethylation could participate in PRMT5 overexpression in the early stages of HCC; however, other regulatory mechanisms might be included in stages II and III, including histone modifications and others. This was clarified in discussion section line 239.

  1. Some figures in the manuscript suffer from issues with legibility and quality. Specifically, the fonts are too small, making the figures difficult to read. Such as Figure 1c, 2e, 2f, 3a, 3c, 4a...

Response: All figures were regenerated to exhibit better resolution and font size.

Reviewer 2 Report

Comments and Suggestions for Authors

PRMT5 Mediated HIF1α Signaling and Ras-Related Nuclear Protein as Promising Biomarker in Hepatocellular Carcinoma examines this arginine methyl transferase as  a marker, along with RAN in a HIF1a mediated pathway. Overall, the study supplies evidence for this with in silico methodology.  However, there are a few problem points. In Fig. 2, different cutoffs for high and low appear to be used, especially notable for panels, A and D vs. B and C. If possible, try and harmonize the criteria for the high vs low division among these K-M analyses. Same for Fig 5 and sup. Fig 4.  Panels E and G seem too good to be true. I note that the description of GSE65485 says there are only 5 normal and that is too small a number for this type of assessment.  Drop panel f and/or find another GSE panel to substitute.  Also, it is hard to see how the ROC derives from data that should be somewhat like that shown in Fig 1, panel a.  The results for RAN seem more likely to produce the ROC results shown in Fig 5 than the PRMT5 set.  Please make it clear that you did the  DN vs HCC work in your lab or not and what portion is from your lab vs. derived from databases. In sup. Fig 1, where is Mapk?  Not clear to me. 

Comments on the Quality of English Language

OK

Author Response

Reviewer 2

PRMT5 Mediated HIF1α Signaling and Ras-Related Nuclear Protein as Promising Biomarker in Hepatocellular Carcinoma examines this arginine methyl transferase as a marker, along with RAN in a HIF1a mediated pathway. Overall, the study supplies evidence for this with in silico methodology.  However, there are a few problem points.

  1. In Fig. 2, different cutoffs for high and low appear to be used, especially notable for panels, A and D vs. B and C. If possible, try and harmonize the criteria for the high vs low division among these K-M analyses. Same for Fig 5 and sup. Fig 4.

Response: Cutoffs for each gene were changed and in harmonized Fig. 2 Fig 5 and sup. Fig 4 and described in the methods section 4.3

  1. Panels E and G seem too good to be true. I note that the description of GSE65485 says there are only 5 normal and that is too small a number for this type of assessment. Drop panel f and/or find another GSE panel to substitute. 

Response: GSE65485 was excluded from the study, and panel f was dropped from fig. 2 and fig. 6 accordingly.

  1. Also, it is hard to see how the ROC derives from data that should be somewhat like that shown in Fig 1, panel a. The results for RAN seem more likely to produce the ROC results shown in Fig 5 than the PRMT5 set.

Response: We re-analyzed the data set after changing the cutoff value from 6 to 5 and we suggest it could be more representative in Fig. 2 panel E.

  1. Please make it clear that you did the DN vs HCC work in your lab or not and what portion is from your lab vs. derived from databases.

Response: Data was obtained from previous publications of the co-authors (Jens-Uwe Marquardt and Darko Castven from Department of Medicine I, University Medical Center Schleswig-Holstein, Campus Lübeck, Lübeck, Germany); this was clarified in the methods section 4.6 line 355. And the previous publication was cited.

  1. In sup. Fig 1, where is Mapk? Not clear to me.

Response: Supplementary Fig 1 represents some genes by corresponding protein names; in this case, MAPK3 is represented as ERK1/2 where ERK1 is an enzyme that in humans is encoded by the MAPK3 gene. This was clarified in line 162.

Round 2

Reviewer 2 Report

Comments and Suggestions for Authors

Much improved.  The science is passable.  The English needs some work--end of second paragraph in section 3- is "crustal" crucial? stands out as one example where some more editing is required.

Comments on the Quality of English Language

Needs some more editing.

Author Response

Reviewer 2

Much improved.  The science is passable.  The English needs some work--end of second paragraph in section 3- is "crustal" crucial? stands out as one example where some more editing is required.

Response: We thank the reviewer for the positive feedback and valuable comments. As per the suggestion, with the help of a native speaker, English language was improved and all the corrections were marked as track changes, including the one identified at the end of second paragraph in section 3 (line 254).